# Rice OsMRG702 and Its Partner OsMRGBP Control Flowering Time through H4 Acetylation

**DOI:** 10.3390/ijms24119219

**Published:** 2023-05-25

**Authors:** Feng Gong, Kaixin Zhang, Jing Wen, Shenbo Yu, Wenjin Li, Gaofeng Du, Cui Wu, Kangjing Zhu, Yifeng Xu

**Affiliations:** College of Life Sciences, Nanjing Agriculture University, Nanjing 210014, China; fgong@njau.edu.cn (F.G.);

**Keywords:** flowering, H4K5ac, OsMRG702, OsMRGBP, rice

## Abstract

MORF-RELATED GENE702 (OsMRG702) regulates flowering time genes in rice, but how it controls transcription is not well known. Here, we found that OsMRGBP can directly interact with OsMRG702. Both *Osmrg702* and *Osmrgbp* mutants show the delayed flowering phenotype with the reduction in the transcription of multiple key flowering time genes, including *Ehd1* and *RFT1*. Chromatin immunoprecipitation study showed that both OsMRG702 and OsMRGBP bind to the *Ehd1* and *RFT1* loci and the absence of either OsMRG702 or OsMRGBP leads to a decrease of H4K5 acetylation at these loci, indicating OsMRG702 and OsMRGBP cooperatively together to promote the H4K5 acetylation. In addition, whilst *Ghd7* are upregulated in both *Osmrg702* and *Osmrgbp* mutants, only OsMRG702 binds to the loci, together with the global increased and *Ghd7* locus-specific increased H4K5ac levels in *Osmrg702* mutants, suggesting an additional negative effect of OsMRG702 on H4K5 acetylation. In summary, OsMRG702 controls flowering gene regulation by altering H4 acetylation in rice; it works either together with OsMRGBP to enhance transcription by promoting H4 acetylation or with other unknown mechanisms to dampen transcription by preventing H4 acetylation.

## 1. Introduction

Flowering is a key biological process in plant reproduction [1]. Many internal and external factors such as temperature, light, hormones, and a series of complexes form molecular regulatory networks affecting flowering. The flowering regulatory network has been thoroughly studied in the dicot model plant *Arabidopsis* and the monocot model plant rice [2]. In *Arabidopsis*, the FLOWERING LOCUS C (FLC) which inhibits flowering and the *FLOWERING LOCUS T* (*FT*) which encodes the florigen are the key regulatory genes in the multiple flowering pathways [3]. The GIGANTEA (GI) promotes the transcription of *CONSTANS* (*CO*) in leaves, and CO directly activates *FT* [4]. Rice is a facultative short-day plant with two florigen genes *RICE FLOWERING LOCUS T1* (*RFT1*) and *HEADING DATE 3A* (*Hd3a*), which are homologous to *FT*. *RFT1* preferentially working on long days, while *Hd3a* mainly works on short days [5,6]. The OsGI-Hd1-Hd3a pathway in rice parallels with the *Arabidopsis* GI-CO-FT pathway, whilst there are also rice-specific components centered on *Early Heading Date 1* (*Ehd1*), *Grain number*, *plant height and heading date* (*Ghd7*), together with *Early Heading Date 3* (*Ehd3*) [7,8,9]. Among these, Ehd3, a plant homeodomain finger-containing protein, is a critical promoter of rice flowering through repressing the floral repressor *Ghd7* [10,11]. Ghd7, a nuclear protein containing the CCT (CONSTANS, CO-LIKE, and TOC1) DOMAINS, is not only the main inhibitor of *Ehd1* under long-day conditions but also an important gene for rice yield and flowering adaptability [12,13]. Ehd1 is a B-type response regulator, which plays a promoting function in flowering [14]. Under long-day conditions, Ehd1 activates *RFT1* expression to accelerate the flowering [14].

Epigenetic regulation, including chromatin remodeling, DNA methylations, histone modifications, and noncoding RNA, can, directly and indirectly, regulate transcription, thus is another important component of the molecular regulatory network of plant flowering [15,16,17]. In general, histone H3K4 trimethylation (H3K4me3), and H3K36 di- and tri-methylation (H3K36me2/me3), together with histone acetylations, were co-related with gene activation, while H3K9 methylation, H3K27 trimethylation (H3K27me3), and histone deacetylation were associated with gene suppression [18]. Among them, histone methylations are mediated by the catalytic activity of histone methyltransferase (HMT) of the set domain group (SDG) proteins, and the resulting changes (e.g., chromatin structure) through the effector complexes (e.g., the histone mark reader containing complexes) affect the transcription of flowering genes [19]. Rice contains 42 SDG proteins; among them, SDG708, SDG724 and its homologous SDG725 promote flowering through promoting the transcription of the flowering genes *Hd3a*, *RFT1*, and *Ehd1* via H3K36 methylation at these loci [20,21,22].

The histone mark reader proteins could mediate downstream events by recognizing the histone marks [23]. In yeast and mammals, the H3K36me3 reader proteins Esa1-associated factor 3 (EAF3) and MORF-RELATED GENE 15 (MRG15) were involved in the nucleosome acetyltransferase of the H4 (NuA4) complex as well as the Trimer Independent of NuA4 involved in transcription interactions with nucleosomes (TINTIN) complex, which regulate transcription either through modifying H4 acetylation level or through controlling transcription elongation process [24,25]. In *Arabidopsis*, the EAF3 homologs MORF-RELATED GENE 1 (MRG1) and MORF-RELATED GENE 2 (MRG2) show divergence in protein sequences but still function redundantly in the transcription regulation of flowering genes [26]. It’s likely that MRG1 and MRG2 recruit the plant histone acetyltransferase HAM1/2 to the H3K36me3 enriched loci to promote H4K5ac, thus maintaining the expression of *FLC* and *FT* [26]. MRG2 may also directly interact with CONSTANS to modulate the *FT* transcription [27]. Recently, it was revealed that the plant-specific histone deacetylase HD2C could compete with CO to bind to MRG2 and modulate the *FT* transcription at dusk [28]. Another study claimed that MRG1 and MRG2 interact with the nucleosome assembly protein family proteins NRP1 and NRP2, to regulate the histone acetylation states of the *FLC* and *FT* loci [29]. In rice, there are two MRG protein family members, OsMRG701 and OsMRG702, which are homologous proteins of *Arabidopsis* MRG1. The two proteins are highly homologous in sequences, and it has been shown that OsMRG702 plays the primary function in regulating flowering genes, mainly through its binding capability to H3K36me3 [30,31]. However, how the OsMRG702 functions in the transcription regulation of rice flowering genes need to be further studied.

Here, we found that OsMRGBP interacts with OsMRG702. Phenotype observation, expression analysis as well as chromatin immunoprecipitation assay revealed that both functions in promoting histone H4 acetylation in the loci of key flowering genes to regulate flowering, whilst OsMRG702 also repressed the transcription of the floral repressor *Ghd7* through preventing H4 acetylation.

## 2. Results

### 2.1. OsMRGBP Directly Interacts with OsMRG702

*OsMRG702* was participated in brassinosteroid-regulated growth and flowering time control in rice, but who is its protein partner and how does it regulate transcription of downstream genes are not well known [30]. In yeast and mammals, OsMRG702 homologs Eaf3 and MRG15 directly interact with Esa1-associated factor 7 (Eaf7) and MRGBP, which contain an eaf7 domain [24,32]. Homologous sequence alignment analysis showed that the eaf7 domain-containing proteins are conserved in land plants (Appendix A). In rice, there is only one Eaf7 homolog protein, we named it OsMRGBP. The “FxLP” motif, which was reported to be essential for Eaf7/MRGBP to interact with Eaf3/MRG15, is conserved in OsMRGBP, suggesting the conservation of the protein interaction between OsMRG702 and OsMRGBP [33] (Figure 1A and Appendix A). *OsMRG702* was reported highly expressed in all tested tissues while *OsMRG701* is barely expressed [30]. Thus, *OsMRG702* was considered a major player in rice growth and development in the MRG protein-containing complexes [30]. We examined the expression pattern of *OsMRGBP*, as well as *OsMRG701* and *OsMRG702*. Consistent with the previous report, the qRT-PCR results showed that both *OsMRG702* and *OsMRGBP* were highly expressed in the root, stem, and leaf, while *OsMRG701* was lowly expressed in these tissues (Figure 1B) [30]. Therefore, we speculate that in most rice tissues, OsMRGBP plays roles mainly through interacting with *OsMRG702*, rather than *OsMRG701*.

To validate the direct interaction between OsMRG702 and OsMRGBP, we have utilized the yeast two-hybrid (Y2H) assay. The Y2H results showed that OsMRGBP and OsMRG702 do interact directly with each other (Figure 1C). Then, we tested the possible interaction of OsMRG702 and OsMRGBP in vivo with the LCA assay, which showed that a strong fluorescent signal could be detected only when both *OsMRG702* and *OsMRGBP* were co-expressed (Figure 1D,E). These results support the conservation of protein interactions between OsMRG702 and MRGBP proteins.

### 2.2. Flowering Time Is Delayed in Osmrg702 and Osmrgbp Mutants

We chose the rice variety Zhonghua 11 (ZH11) as the main material in this study for its high Agrobacterium-mediated transformation efficiency and that it is adapted to planting in Nanjing [34]. OsMRG702 knockdown rice showed a relatively late flowering phenotype in the study of OsMRG702 RNAi in the similar Nipponbare variety [30]. To further test whether OsMRGBP is also participating in flowering-time regulation in rice as OsMRG702, the CRISPR/Cas9 technology was applied to obtain *Osmrgbp* and *Osmrg702* mutants for analysis. The first exon of each gene was picked as the target region to design sgRNAs (Figure 2A,B). After resistance screening and sequencing, several mutants were obtained, lines containing one base insertion at the exons in *OsMRG702* and *OsMRGBP* among the mutants were picked out and named as *Osmrg702* and *Osmrgbp* (Appendix A and Figure 2A,B). The *Osmrg702* and *Osmrgbp* mutants flowered around 11–18 days later than that of WT under natural long-day (NLD) conditions in the field (Figure 2C,D). The rachis branches of *Osmrgbp* and *Osmrg702* were fewer than the wild-type (Figure 2E). To test whether the *Osmrg702* and *Osmrgbp* mutation affects grain yield and yield components, we evaluated the agronomic traits, including several tillers and hundred-grain weight in WT, *Osmrg702*, and *Osmrgbp* grown in the paddy field under NLD conditions. The number of tillers and hundred-grain weight in both mutants were decreased compared with these of WT and the defects in *Osmrgbp* are relatively severe (Figure 2F,G).

### 2.3. Multiple Flowering Genes Are Altered in Expression Levels in Osmrg702 and Osmrgbp Mutants

It has been reported that rice florigen genes *Hd3a* and *RFT1*, together with their upstream regulators *Ehd3* and *Ehd1*, are down-regulated in *OsMRG702* RNAi knockdown rice in both LD and SD conditions [30]. Consistent with this report, we observed the significant reduction in transcripts of *Hd3a*, *RFT1*, *Ehd3*, and *Ehd1* in both *Osmrg702* and *Osmrgbp* mutants in the LD conditions (Figure 3A,B). The reduction is relatively severe in *Ehd1*, *RFT1*, and *Hd3a*, even the transcription of *Hd3a* is at very low level (100 fold less compared with that of *RFT1*) under the LD conditions (Figure 3B). Compared with the data with the *OsMRG702* RNAi knockdown plants, the defect in transcription level is much stronger in *Ehd1*, *RFT1*, and *Hd3a* (Figure 3B) [32]. In contrast, the *Ehd3* gene, which is the major *Ghd7* repressor, is slightly reduced in *Osmrg702* and *Osmrgbp* mutants, which are comparable with that in the *OsMRG702* RNAi rice (Figure 3B) [32]. However, the effect of the *Osmrgbp* mutation is slightly stronger than that of *Osmrg702* (Figure 3B). Interestingly, the expression of *Ghd7* was upregulated in both mutants (Figure 3B), which before is not reported in *OsMRG702* RNAi; instead, it was reported to be increased in *SDG725* RNAi plants [30]. In summary, we concluded that both OsMRG702 and OsMRGBP promote rice flowering by activating flowering promoter genes and repressing the floral repressor gene *Ghd7*.

### 2.4. OsMRG702 and OsMRGBP Directly Bind to the Loci of Multiple Key Flowering Genes

To examine how OsMRG702 and OsMRGBP regulate these flowering genes during the LD conditions, we prepared the transgenic plants overexpressing *OsMRG702-3Flag* and *OSMRGBP-3HA*. After testing with qRT-PCR and Western blots, two individual lines for each construct were chosen for further study (Appendix A). We chose the loci around the transcription start site (TSS) of the tested genes, where H3K36me3 enriched, for ChIP-qPCR analysis, as OsMRG702 was reported to recognize H3K36me3 [35]. The chromatin immunoprecipitation assay showed that OsMRG702 and OSMRGBP were enriched at the promoter region in the *Ehd1* loci, and at the TSS region in the *RFT1* and *Ehd3* loci (Figure 4A–C). However, neither OsMRG702 nor OsMRGBP binds to *Hd3a*. The very low expression level of *Hd3a* in the LD conditions implies the possible lowly binding of OsMRG702 and OsMRGBP. It is noticed that OsMRG702 and OsMRGBP bind nearly almost the same loci of the tested genes, suggesting that OsMRG702 and OsMRGBP may coordinate together in regulating *Ehd1*, *RFT1*, and *Ehd3* expression. It is noteworthy that only OsMRG702 but not OsMRGBP, binds to the *Ghd7* loci, implying an OsMRGBP-independent function of OsMRG702 at certain loci (Figure 4D).

### 2.5. The H4K5ac Level of the Key Flowering Time Genes Were Decreased in Osmrg702 and Osmrgbp

In yeast and mammals, OsMRG702 and OsMRGBP homologs participated in the NuA4 complex as well as the TINTIN complex, regulating transcription through modifying the H4 acetylation level or transcription elongation process [24,25]. *Arabidopsis* MRG1/2, the homologs of OsMRG702, interact with the histone H4-specific acetyltransferases HAM1/2 and promote H4 acetylation level in flowering genes *FT* and *FLC* to modulate their transcriptions [26]. Thus, we performed a ChIP-qPCR assay to examine the H4K5 acetylation level at the rice flowering time genes. The result showed that the H4K5ac levels at *Ehd1* loci were significantly decreased in the promoter region and the TSS region in both mutants compared with WT at 60 days after germination, and no significant difference between *Osmrg702* and *Osmrgbp* was found except the site 4 region (Figure 5A). H4K5ac levels at the *Ehd3* and *RFT1* loci were similarly decreased in both mutants (Figure 5B,C). At the *Ghd7* loci, H4K5ac levels were increased only in *Osmrg702*, while there is no difference in *Osmrgbp*, which is consistent with the different binding profiles of OsMRG702 and OsMRGBP at *Ghd7* (Figure 4D and Figure 5D). Thus, either OsMRG702 or OsMRGBP is essential for the high level of H4 acetylation of their targeted flowering genes. They may function coordinately in promoting H4 acetylation, possibly through recruiting the H4K5 histone acetyltransferase to their binding loci. OsMRG702 has the additional reverse function in H4K5 acetylation in some loci, e.g., the Ghd7 loci, which is OsMRGBP-independent.

## 3. Discussion

OsMORF-RELATED GENE702 is a reader protein of trimethylated histone H3 Lysine 4 and histone H3 lysine 36 and is involved in flowering time control in rice [30]. How it regulates flowering genes were not well understood. Identifying and characterizing its protein partner could help us to elucidate its molecular mechanism. The homologs of OsMRGBP in yeast and mammalian can directly interact with OsMRG702 homologs through the FxLP motif. In this study, we found a protein OsMRGBP (protein homologous to human MRGBP) that interacts with OsMRG702 in rice, which also contains the conserved FxLP motif. Moreover, both the loss-of-function mutants of *Osmrg702* and *Osmrgbp* showed late flowering phenotypes under the LD conditions. Expression analysis showed that multiple flowering genes are downregulated significantly in both *Osmrg702* and *Osmrgbp*, including the florigen gene *RFT1* and *Hd3a*, and the rice-specific flowering gene *Ehd1*, which is consistent with the reported OsMRG702 RNAi knockdown lines [30]. At the same time, the ChIP-qPCR assay showed that both OsMRG702 and OsMRGBP directly bind to these genes, suggesting that transcription reduction is the direct consequence of *OsMRG702*/*OsMRGBP* mutation. It is noteworthy that OsMRG702 and OsMRGBP bind similar regions of the genes, suggesting OsMRG702 and OsMRGBP could function coordinately in transcription upregulation. Based on these results, we conclude that OsMRGBP cooperates with OsMRG702 and directly activates flowering regulatory genes to promote rice flowering.

In yeast and human, the OsMRG702 homologs (Eaf3/MRG15) and the OsMRGBP homolog (Eaf7/MRGBP)participate in both the NuA4/Tip60 complex and the TINTIN complex, to promote gene transcription through maintaining/increasing H4 acetylation level and to regulate the transcriptional process, respectively [24,33]. It is reported that *Arabidopsis* MRG1 and MRG2, the two functional redundant homologs of OsMRG702, recruit the NuA4 components, the H4-specific acetyltransferases HAM1/2, to maintain/increase H4 acetylation levels at the loci of the floral repressor *FLC* and its downstream florigen gene *FT* [26]. In rice, ChIP-qPCR assay showed that the H4K5ac levels are decreased in the *Ehd1* and *RFT1* promoter and TSS regions in *Osmrg702* and *Osmrgbp* mutants, suggesting that OsMRG702 and OsMRGBP are likely to regulate H4 acetylation to maintain or increase the transcription of *Ehd1* and *RFT1*. Considering that the binding sites of OsMRG702 and OsMRGBP are similar at the gene loci of both *Ehd1* and *RFT1*, implying that OsMRG702 and OsMRGBP work together in H4K5 acetylation modulating, possibly function as the rice NuA4 subunits. OsYAF9, another subunit of the NuA4 complex in rice, was also reported to be responsible for H4 acetylation. *Osyaf9* mutants showed the phenotype with late flowering, reduced height, fewer tillers, and fewer pollen grains [36]. The similar phenotypes among *Osyaf9*, *Osmrg702*, and *Osmrgbp* mutants suggest that the NuA4-mediated H4 acetylation mainly controls the development of rice during reproduction.

In contrast to the above flowering time genes, the *Ghd7* transcription was increased significantly in both *Osmrg702* and *Osmrgbp* mutants. However, the H4K5ac level at the *Ghd7* loci is increased only in the *Osmrg702* mutant, whilst there is no change in *Osmrgbp* when compared with that in WT. As shown in Figure 4D, only OsMRG702 can bind to the *Ghd7* loci, suggesting that the upregulation of *Ghd7* in *Osmrgbp* is an indirect effect of OsMRGBP, e.g., the reduction of the upstream repressor *Ehd3*. It has been reported that *Arabidopsis* MRG2 could interact with plant-specific histone deacetylase HD2C and function in histone deacetylation at the *FT* loci [28]. Thus, it is likely that OsMRG702 is conserved in the interaction with rice HD2C homologs and plays a role in the H4 deacetylation as well. Western blot analysis of the global H4K5ac level in *Osmrg702* and *Osmrgbp* strongly supports the idea that *Osmrg702* functions in the histone H4K5 deacetylation (Appendix A). Thus, OsMRG702 modulates flowering time by regulating the H4K5ac level at the rice key flowering genes both positively with OsMRGBP and negatively with other partners, e.g., HD2C, which requires further studies.

## 4. Materials and Methods

### 4.1. Plant Materials and Growth Conditions

Rice (Zhonghua11, ZH11) plants were used in this study. For the phenotype study, plants were grown in Nanjing, under a natural LD photoperiod. The plants used for molecular experiments were grown in growth chambers under an LD photoperiod (14 h of light and 10 h of dark at 28 °C).

### 4.2. CRISPR/Cas9 for Genome Editing

The single guide RNA (sgRNA) for the CRISPR/Cas9 system was designed at http://cbi.hzau.edu.cn/cgi-bin/CRISPR2/SCORE (CRISPR-P 2.0, accessed on 12 June 2019). Two 20-bp target sequences, 5′-ACTCTCTCGCCCTCCTTGAA-3′ and 5′-CCGCAGCTCCAGCTCCACCT-3′, which were in the front part of the coding sequence were picked up for *OsMRG702* and *OsMRGBP*, respectively. CRISPR/Cas9 vector construction kit (Biogle, Changzhou, China) was used to generate the Oligo dimer and ligate the Oligo dimer to the Cas9 destination vector. These constructs were introduced into the Agrobacterium strain EHA105. Callus was induced from mature embryos of rice cultivar ZH11 for Agrobacterium-mediated rice transformation [37]. Transgenic lines were screened with hygromycin resistance, and the mutants were identified through PCR amplification followed by sequencing. Genomic sequences of *OsMRG702* and *OsMRGBP* were obtained from Phytozome (https://phytozome.jgi.doe.gov/, accessed on 1 May 2019).

### 4.3. Plasmid Construction

To generate *OsMRG702* or *OsMRGBP* overexpression lines, the constructs of *pCAMBIA1301-35S::OsMRG702-3Flag* and *pCAMBIA1301-35S::OsMRGBP-3HA* were prepared as follows: coding sequence of *OsMRG702* and *OsMRGBP* fusion with 3xFlag or 3xHA tag were cloned with primer sets 1301-35S::OsMRG702_F and 1301-35S::OsMRG702_R, 1301-35S::OsMRGBP_F and 1301-35S::OsMRGBP_R, 3Flag_F and 3Flag_R, and 3HA_F and 3HA_R, respectively. Then, the PCR fragments were cloned into the vector pCAMBIA1301 in the restriction sites HindШ and BstEП through homologous recombination using the Uniclone One Step Seamless Cloning Kit (Genesand, Beijing, China). Then, each vector was introduced into ZH11 with Agrobacteria-mediated transformation as described in the section for CRISPR/Cas9 system-based genome editing. Transgenic lines were screened with hygromycin resistance and were verified with qRT-PCR as well as Western blot.

For yeast two-hybrid assay, the *OsMRG702* and *OsMRGBP* coding sequences were amplified with primer sets AD-OsMRG702_F and AD-OsMRG702_R, AD-OsMRGBP_F and AD-OsMRGBP_R, BD-OsMRG702_F and BD-OsMRG702_R, and BD-OsMRGBP_F and BD-OsMRGBP_R, respectively. Then, each fragment was cloned individually into pGADT7 and pGBKT7 both digested by EcoRІ through homologous recombination with the kit (Genesand, Beijing, China) as above.

For luciferase complementation assay, the *OsMRG702* and *OsMRGBP* coding sequence, which was amplified by primer set cLUC-OsMRG702_F and cLUC-OsMRG702_R, cLUC-OsMRGBP_F and cLUC-OsMRGBP_R, nLUC-OsMRG702_F and nLUC-OsMRG702_R, nLUC-OsMRGBP_F and nLUC-OsMRGBP_R, were cloned into both pCambia1300-cLUC and pCambia1300-nLUC digested by BamHІ and SalІ through homologous recombination as above. All primers were listed in Appendix A.

### 4.4. Yeast Two-Hybrid (Y2H)

pGADT7 and pGBKT7 constructs carrying target genes were transformed into yeast AH109 and YH187 competent cells and streaked at SD-Leu or SD-Trp plate, respectively. Positive colonies from each transformed strain were picked for co-culturing and mating in YPDA liquid medium at 30 °C overnight, then the interaction between the bait and prey was observed on SD-Leu/-Trp solid medium and SD-His/-Leu/-Trp/-Ade solid medium. Plates were incubated for 4 days at 30 °C.

### 4.5. Luciferase Complementation Assay (LCA)

The plasmids containing cLUC and nLUC pairs were transformed into Agrobacterium. The monoclonal was picked up and cultured overnight in an LB liquid medium with antibiotic selection. After centrifugation, the pellet was resuspended with the resuspension buffer (100 mM MES, 1 M MgCl_2,_ and 150 mM acetosyringone), and adjusted the OD_600_ to 0.4. After incubation for 3 h at room temperature, the suspension was infiltrated into *N. benthamiana* leaves with a 1 mL syringe. Two days later, leaf discs were taken and incubated with 100 mM luciferin for 5 min; the Tanon-5100 Multi system was applied to take the luminescence images.

### 4.6. Quantitative RT-PCR

Leaf blades of 60-day-old plants grown in the LD growth chamber were harvested at ZT2 to extract total RNA. The RNA was extracted using Trizol Reagent according to the manufacturer’s instructions (http://www.monadbiotech.com/, accessed on 4 May 2021). After RNA quantification, cDNA was synthesized with 1 µg of total RNA by 4 × gDNA wiper Mix and Oligo(dT)23 (Vazyme, Nanjing, China) at 42 °C for 2 min. After adding 5 × HiScript П Select qRT SuperMix П (Vazyme, Nanjing, China) at 55 °C for 15 min and 85 °C for 5 s. Quantitative RT-PCR analysis was performed using 2 × AceQ Universal SYBR qPCR Master Mix (Vazyme, Nanjing, Jiangsu, China) on a QuantStudio^TM^ 1 Real-Time PCR Instrument (96-well 0.2 mL Block) (Yeasen, Shanghai, China). Each experiment was repeated three times and samples were normalized by OsACTIN10-1. Data analyses were performed with GraphPad Prism software; the relative expression levels were measured using the 2 (−∆∆Ct) analysis method and the error bars in the figures represent the variance of three biological replicates.

### 4.7. Chromatin Immunoprecipitation (ChIP)

Leaf blades of 60-day-old plants grown in the LD growth chamber were harvested at ZT2 (Two hours after dawn). ChIP assays were performed as described previously using the following antibodies: anti-acetyl-H4K5 (Abcam, Waltham, MA, USA), anti-Flag (Smart-Lifesciences, Changzhou, China), and anti-HA (Smart-Lifesciences, Changzhou, China) antibodies. DNA fragments were recovered by DNA Extraction Reagent (Cat#P1012; shkxbio, Beijing, China) [26]. Finally, the precipitated DNA was quantified by qPCR with the primers listed in Appendix A.

### 4.8. Nuclear Extractions and Western Blot

Plant materials were ground in liquid nitrogen, and then extracted in NEB1 buffer (0.4 M sucrose, 10 mM PH8.0 Tris-HCl, 5 mM β-Mercaptoethanol, 0.1 mM PMSF, 0.1 mM Protease Inhibitor (Yeasen, Shanghai, China) on ice. Then, the crude extracts were filtered through the Miracloth (Millipore, Madison, WI, USA), and the supernatant was gathered into a 50 mL falcon tube. The nuclei were pelleted by centrifugation at 10,000× *g* for 5 min at 4 °C and washed two times with NEB2 buffer (0.25 M sucrose, 10 mM Tris-HCl PH-8.0, 10 mM MgCl_2_, 1% Triton X-100, 5 mM β-Mercaptoethanol, 0.1 mM PMSF, 0.1 mM Protease Inhibitor). The samples were centrifuged at 12,000× *g* for 10 min at 4 °C after each washing. The pellets were resuspended by NEB3 buffer (1.7 M sucrose, 10 mM PH8.0 Tris-HCl, 2 mM MgCl_2_, 0.15% Triton X-100, 5 mM β-Mercaptoethanol, 0.1 mM PMSF, 0.1 mM Protease Inhibitor) and centrifuge at 13,000 rpm for 1 h at 4 °C. The nuclear pellets were reserved and resuspended with SDS-lysis buffer (1 M NaHCO_3_, 10% SDS) on ice for 30 min. Then, the suspension was diluted with dilution buffer (0.01% SDS, 1.1% Triton X-100, 1.2 mM EDTA, 16.7 mM Tris-HCl, 167 mM NaCl, 0.1 mM PMSF, 0.1 mM Protease Inhibitor) and mixed well with gentle inversion. After centrifugation at 15,000 rpm at 4 °C for 10 min, the supernatant was collected and heated at 95 °C for 15 min after adding 5 × SDS-PAGE protein loading buffer (Yeasen, Shanghai, China). Then, the samples were analyzed by Western Blot with anti-acetyl-H4K5 (Abcam, Waltham, MA, USA). Histone H3 was used as a loading control for nuclear protein with an anti-H3 antibody (Abcam, Waltham, MA, USA).

### 4.9. Measurement of Agronomic Traits

To investigate agronomic traits, WT, *Osmrg702*, and *Osmrgbp* plants were grown in the paddy field under NLD conditions in Nanjing. Days to flowering and the number of tillers were measured just after the heading. Hundred-grain weights were examined after harvest.

### 4.10. Phylogenetic Analysis

The sequences of eaf7 domain-containing proteins in plants that represent different evolutionary stages were obtained through blast with *yeast* Eaf7. These plant sequences and some previously identified eaf7 domains from yeast and animals were aligned using ClustalW. The phylogenetic tree was constructed using MEGA5.05 software with bootstrapping set at 1000 replicates.

### 4.11. Accession Numbers

Sequence data from this article can be found in the Rice Annotation Project (https://rapdb.dna.affrc.go.jp/, accessed on 4 April 2020.) under numbers Os04g01130 (OsMRG701), Os11g34300 (OsMRG702) and Os10g051000 (OsACTIN10), Os05g0512500 (OsMRGBP), Os06g0298200 (Hd1), Os08g0105000 (Ehd3), Os07g0261200 (Ghd7), Os10g0463400 (Ehd1), Os06g0157500 (RFT1) and 0s06g0157700 (Hd3a).

## Figures and Tables

**Figure 1 ijms-24-09219-f001:**
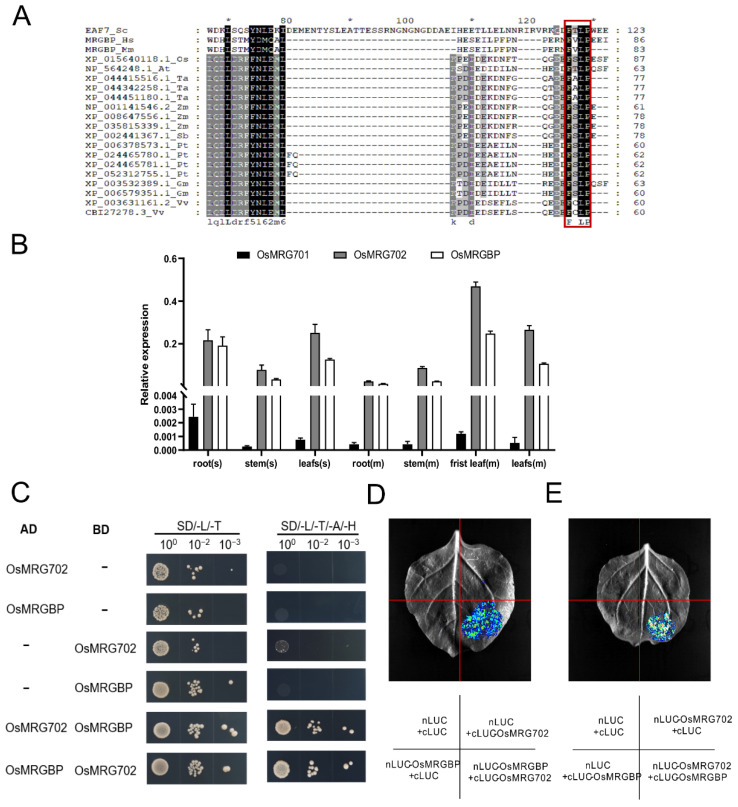
OsMRG702 interacts with OsMRGBP. (**A**) Alignment of full-length sequences of MRGBP proteins from human (Hs), mouse (Mm), *Oryza sativa L.* (Os), *Arabidopsis thaliana* (At), *Triticum aestivum L.* (Ta), *Zea mays L*. (Zm), *Sorghum Moench* (Sb), *Populus trichocarpa* (Pt), *Glycine max* (Gm), *Vitis vinifera* (Vv), in which the conserved FxLP motif is highlighted by red box. Amino acid sequence numbers are marked by * and numbers. (**B**) Relative expression levels of OsMRG701, OsMRG702, and OsMRGBP in different rice tissues. s, seedlings. m, mature plants. (**C**) The Y2H assay to validate OsMRG702 interacting with OsMRGBP. (**D**,**E**) LCA to verify the interaction between OsMRG702 and OsMRGBP in vivo.

**Figure 2 ijms-24-09219-f002:**
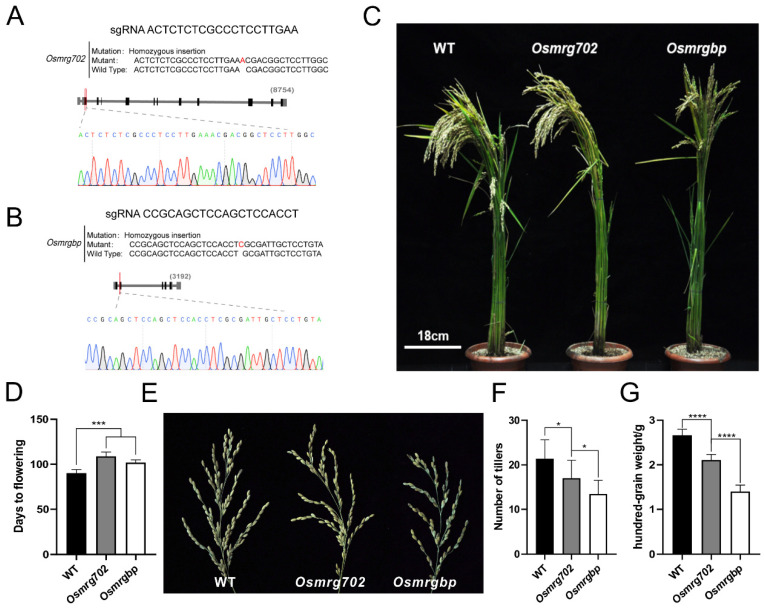
Flowering time is delayed in *Osmrg702* and *Osmrgbp* mutants. (**A**,**B**) The sgRNA sequences and the mutated sequences of CRISPR/Cas9-based *Osmrg702* (**A**) and *Osmrgbp* mutants (**B**). (**C**) Plant morphologies of the WT, *Osmrg702*, and *Osmrgbp* show slightly later flowering phenotypes of *Osmrg702* and *Osmrgbp* in NLD conditions. (**D**) Statistic analysis of flowering times of the WT, *Osmrg702*, and *Osmrgbp* plants based on the days after germination. (**E**) Spikes show fewer rachis branches in *Osmrg702* and *Osmrgbp* plants compared with that of WT. (**F**,**G**) Agronomic trait analysis shows the reduction of a number of tillers and hundred-grain weight in *Osmrg702* and *Osmrgbp* mutants and the latter with stronger defects. Asterisks indicate statistically significant differences in Student’s *t*-tests. * means *p* < 0.05, *** means *p* < 0.001, **** means *p* < 0.0001.

**Figure 3 ijms-24-09219-f003:**
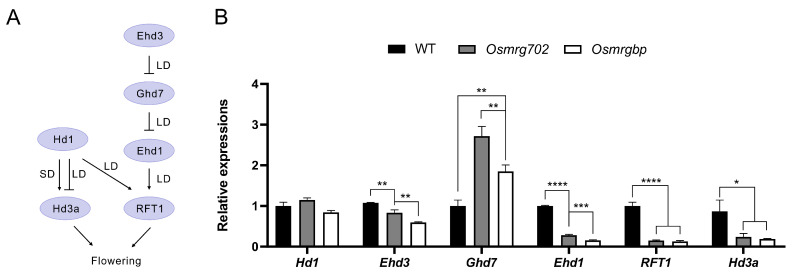
Multiple flowering regulatory genes are down-regulated in the mutants. (**A**) Schematic of core flowering regulatory pathways in rice. (**B**) Relative transcript levels of flowering regulatory genes in the WT, *Osmrg702*, and *Osmrgbp* plants. The relative expression levels of *Hd3a* were indicated by the right *Y*-axis while the rest tested genes were indicated by the left *Y*-axis. Asterisks indicate statistically significant differences in Student’s *t*-tests. * means *p* < 0.05, ** means *p* < 0.01, *** means *p* < 0.001, **** means *p* < 0.0001.

**Figure 4 ijms-24-09219-f004:**
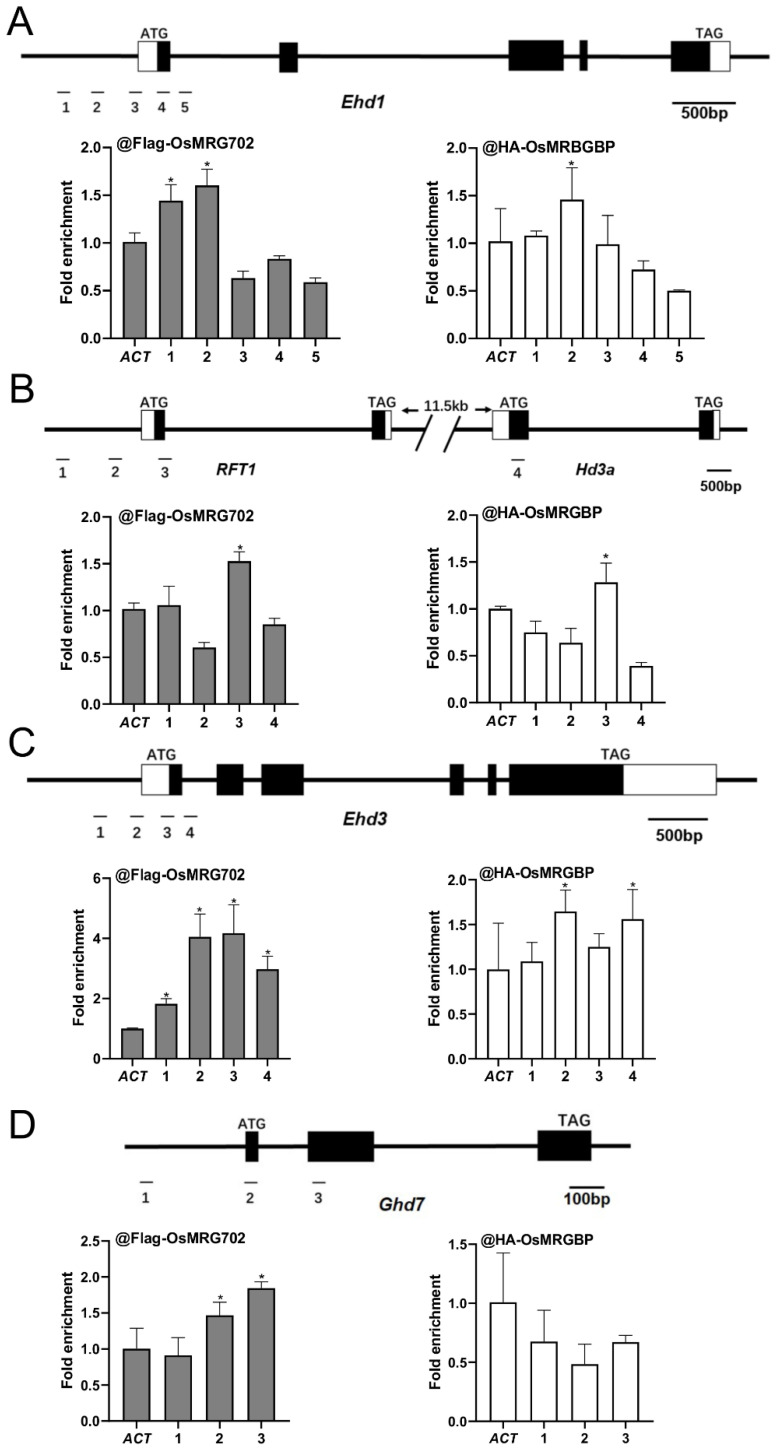
OsMRG702 and OsMRGBP enrichment at the loci of key flowering genes. OsMRG702 and OsMRGBP enrichment at the *Ehd1* (**A**), *RFT1* and *Hd3a* (**B**), *Ehd3* (**C**), and *Ghd7* (**D**) loci were analyzed with ChIP-qPCR. The enrichment on the OsACTIN-2 (ACT) locus was normalized to 1. Asterisks indicate statistically significant differences of Student’s *t*-test between the tested loci and ACT. * means *p* < 0.05.

**Figure 5 ijms-24-09219-f005:**
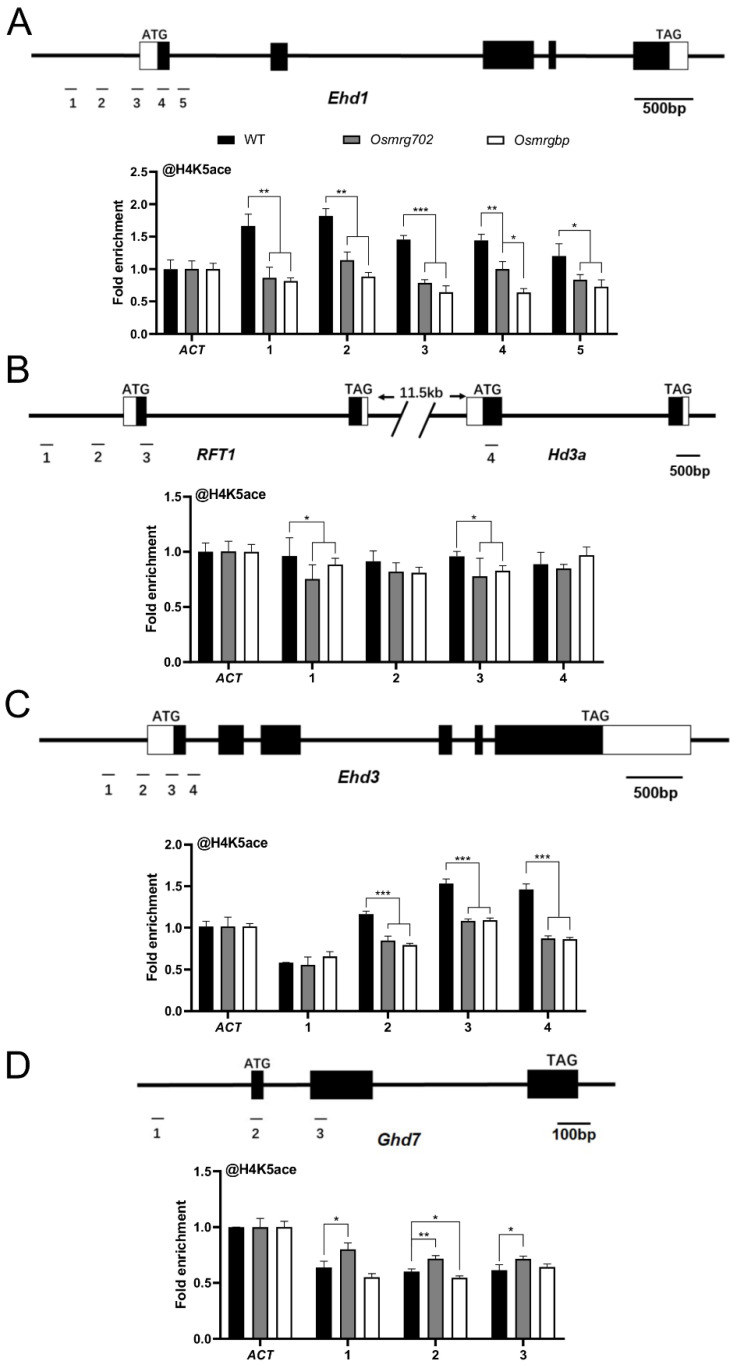
H4K5 acetylation enrichment at the loci of key flowering genes. ChIP-qPCR analysis presenting the enrichment levels of H4K5 acetylation of the *Ehd1* (**A**), *RFT1* and *Hd3a* (**B**), *Ehd3* (**C**), and *Ghd7* (**D**) loci. The enrichment on the ACT locus was normalized to 1. Asterisks indicate statistically significant differences of Student’s *t*-test between WT and mutants at the tested locus. * means *p* < 0.05, ** means *p* < 0.01, *** means *p* < 0.001.

## Data Availability

All data generated or analyzed during this study are included in this article and its Appendix A.

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
