# Peer review of "Rice OsMRG702 and Its Partner OsMRGBP Control Flowering Time through H4 Acetylation"

_ijms, 2023, doi:10.3390/ijms24119219_

Round 1
Reviewer 1 Report
Comments and Suggestions for Authors
I have now had the opportunity to examine your manuscript, and I enjoyed reading it. I have read your manuscript, "Rice OsMRG702 and its partner OsMRGBP Control Flowering Time through H4 Acetylation".
My minor comments are (Suggestions to improve):
1- Introduction needs to be improved, indicating the scientific contribution to OsMRG702. In this regard, the literature review needs to be improved
2- Improve the Introduction using recent publications.
3- Improve the discussion using recent publications
4- Where?? and use more descriptive and precise expressions in the conclusion section
5- 2 g, 5 g, 3 percentage - 4 °C, 2-3 days ......... Kindly change to two, five, three............
6- Please, explain why did you use this material.
7- Check the English spelling; and structure throughout the manuscripts
8- Inline 257 and 104, Kindly change but to In contrast.
9- The discussion is not elaborated.

Author Response
I have now had the opportunity to examine your manuscript, and I enjoyed reading it. I have read your manuscript, "Rice OsMRG702 and its partner OsMRGBP Control Flowering Time through H4 Acetylation".
Response: We feel great thanks for your professional review work on our article. According to your nice suggestions, we have made extensive corrections to our previous draft, the detailed corrections are listed below.
My minor comments are (Suggestions to improve):
- Introduction needs to be improved, indicating the scientific contribution to OsMRG702. In this regard, the literature review needs to be improved
Response: Thanks for your suggestion. We have modified the introduction carefully and added several recent and important literatures about OsMRG702. Please read thought the introduction part in the revised manuscript with track change.
- Improve the Introduction using recent publications.
Response: Thanks for your suggestion. We have renewed the introduction part with more recent publications about epigenetics.
- Improve the discussion using recent publications
Response: Thanks for your suggestion. We have added the OsYAF9 rice paper in the part of discussion.
- Where?? and use more descriptive and precise expressions in the conclusion section
Response: Thanks for your suggestion. We have tried our best to polish the language in the revised manuscript.
- 2 g, 5 g, 3 percentage - 4 °C, 2-3 days ......... Kindly change to two, five, three............
Response: Thanks for your suggestion. We have checked the arabic numbers at the beginning to English expression in the revised manuscript.
- Please, explain why did you use this material.
Response: Compared to the RNAi technology, CRISPR/Cas9 technology can be applied to obtain the total-of-function mutants for characterization.
- Check the English spelling;and structure throughout the manuscripts
Response: Thanks for your suggestion. Based on your comments, we have carefully revised the whole manuscripts
- In line 257 and104, Kindly change but to In
Response: Thanks for your suggestion. We have corrected the sentence.
- The discussion is not elaborated.
Response: Thanks for your suggestion. We have corrected the content in discussion and made it easier to follow up.
Reviewer 2 Report
In this manuscript, Gong et al., investigated the function of OsMRG702 and its interacting partner OsMRGBP in flowering time control. They found that Loss of OsMRG702 or OsMRGBP led to reduced expression of RFT1 and Ehd1, which is consistent with the flowering phenotype of Osmrg702 and Osmrgbp mutants. In addition, the authors demonstrate that OsMRG702 and OsMRGBP directly bind to Ehd3, Ehd1, and RFT1 loci and are required for the enrichment of H4K5ac at their loci. Overall, this study provides new insights into the regulatory mechanism of OsMRG702 and OsMRGBP in gene transcription control. Below please find my suggestions.
1. Some figure panels do not appear in sequence in the text. For instance, figure 1B is cited after 1D, figure 2D is cited after 2E.
2. Line 97-99, please provide reference(s) here.
3. Line 114-115, please provide reference(s) here.
4. Line 126, the manuscript does not show the plant height data.
5. Line 130, the manuscript does not present any data to show the reduction in spikelet fertility, does the reduced rachis branches mean reduced fertility?
6. Line 152, please provide reference(s) for the SDG725 RNAi plants.
7. Do OsMRG702 and OsMRGBP overexpression plants show any developmental phenotypes?
8. Line 196, based on the results provided, here “increased” should be “decreased”.
